# Lineage overwhelms environmental conditions in determining rhizosphere bacterial community structure in a cosmopolitan invasive plant

Jennifer L. Bowen[1], Patrick J. Kearns[1], Jarrett E.K. Byrnes[2], Sara Wigginton[3], Warwick J. Allen[4,5], Michael Greenwood[2], Khang Tran[2], Jennifer Yu[2], James T. Cronin[4] & Laura A. Meyerson[3]

Plant–microbe interactions play crucial roles in species invasions but are rarely investigated at the intraspecific level. Here, we study these interactions in three lineages of a globally distributed plant, *Phragmites australis*. We use field surveys and a common garden experiment to analyze bacterial communities in the rhizosphere of *P. australis* stands from native, introduced, and Gulf lineages to determine lineage-specific controls on rhizosphere bacteria. We show that within-lineage bacterial communities are similar, but are distinct among lineages, which is consistent with our results in a complementary common garden experiment. Introduced *P. australis* rhizosphere bacterial communities have lower abundances of pathways involved in antimicrobial biosynthesis and degradation, suggesting a lower exposure to enemy attack than native and Gulf lineages. However, lineage and not rhizosphere bacterial communities dictate individual plant growth in the common garden experiment. We conclude that lineage is crucial for determination of both rhizosphere bacterial communities and plant fitness.

[1] Department of Marine and Environmental Sciences, Marine Science Center, Northeastern University, 430 Nahant Road, Nahant, MA 01908, USA. [2] University of Massachusetts Boston, 100 Morrissey Boulevard, Boston, MA 02125, USA. [3] Department of Natural Resources Science, University of Rhode Island, Woodward Hall, 9 East Alumni Avenue, Kingston, RI 02881, USA. [4] Department of Biological Sciences, Louisiana State University, 202 Life Sciences Building, Baton Rouge, LA 70803, USA. [5] Present address: The Bio-Protection Research Centre, Lincoln University, PO Box 84, Lincoln 7647, New Zealand. Correspondence and requests for materials should be addressed to J.L.B. (email: je.bowen@northeastern.edu)

Interactions between plants and soil microorganisms that occur in the rhizosphere create microhabitats that result in plant species-specific rhizosphere microbial communities. These interactions are highly localized in the region immediately adjacent to plant roots and play a critical role in plant fitness[1, 2]. There is extensive literature on host–microbe interactions[3, 4], much of which suggests that environmental factors typically outweigh heritable factors in structuring microbiomes[5], despite genome wide association studies that link host loci to microbial community structure[6]. However, we lack an understanding of the extent of intraspecific variation in plant–soil microbe interactions, particularly among invasive plant populations, which could provide insights into why some genotypes are more successful than others. While there is evidence of intraspecific variation in microbial communities among cultivated plant species[7–12], it is unclear whether intraspecific forcing is sufficient to overwhelm environmental influences to structure rhizosphere microbial communities in wild plant species.

Micallef et al.[8] documented unique rhizosphere microbial communities among different genotypes of *Arabidopsis thaliana* when grown in common soil and attributed those changes to genotype-specific variation in root exudates. Similarly, different wheat[9], corn[10, 11], and rice[12] cultivars all demonstrate unique microbial signatures in their rhizosphere soils. However, most studies of this nature are typically executed either in a laboratory setting with common soil or in a small number of similar field plots[9]. Exceptions, however, include a study of corn cultivars grown across multiple locations that suggests geographical differences are more important than cultivar differences in structuring microbial communities[7], a pattern that is duplicated in natural populations of the cottonwood tree (*Populus deltoides*)[13]. Very few studies have evaluated the interplay between biogeography and intraspecific genetic factors in structuring the rhizosphere microbiome.

Refining our understanding of the interactions between plant genotype, biogeography, and the microbiome is a critically important research priority to promote sustainable agriculture[14] and is also crucial to our understanding of plant invasions. Escape from native soil plant pathogens is often invoked as one reason for the success of exotic invasive species[15]. Reduced pathogen loads could decrease the need of invaders to allocate resources to pathogen defense, for example via root exudates that stimulate antibiotic production in the rhizosphere[8]. These resources, instead, could be reallocated to traits that enhance the competitive ability of the invasive plant[16]. If true, we might expect to see a greater homogeneity in the microbial community structure of native populations compared to introduced populations because of the need to maintain robust pathogen defenses. In contrast, invasive populations that do not use resources for pathogen defense would be more likely to have divergent microbial communities that are dominated by local soil generalists[16], therefore showing decreasing similarity with distance among invasive lineages.

*Phragmites australis* is a globally distributed plant that is also a harmful invader in North America. It is among the best-studied uncultivated plants[17] and is considered a model species for plant invasions[18]. In North America there are at least three lineages of *P. australis* that grow sympatrically, including an invasive lineage (haplotype M, hereafter "Introduced") that was introduced to North America from Europe[19] at least 150 years ago[20]. Other lineages of *P. australis* include multiple haplotypes of the native North American lineage (hereafter "Native"), and one Gulf coast lineage, (haplotype I, hereafter "Gulf"), whose origin is undetermined but appears to have been present in North America for thousands of years[19–26]. Across North America, Introduced *P. australis* is highly dominant in coastal and inland marshes, affects multiple trophic levels and ecosystem processes, and is costly to eradicate[27]. Because the three different lineages of *P. australis* are widespread and grow sympatrically across North America, this species provides a unique opportunity to study the similarities and differences in the rhizosphere bacterial communities both among plant lineages with known phylogenies and across a wide range of geographical distances.

Here, we investigated whether there were distinct rhizosphere bacterial communities associated with different *P. australis* lineages in both a field survey and common garden experiment. We hypothesized that if plant–microbe interactions facilitated invasions, Introduced *P. australis* populations would have more heterogeneous rhizosphere bacterial communities than Native or Gulf populations. Since soil microbes are typically controlled by local edaphic characteristics[28–31], homogeneity of rhizosphere bacteria within *P. australis* lineages across the sampling range, along with heterogeneity among lineages in our field experiment, would be consistent with our hypothesis that lineage-specific forcing is structuring rhizosphere bacterial communities. Our results, confirmed in both a field survey and a common garden

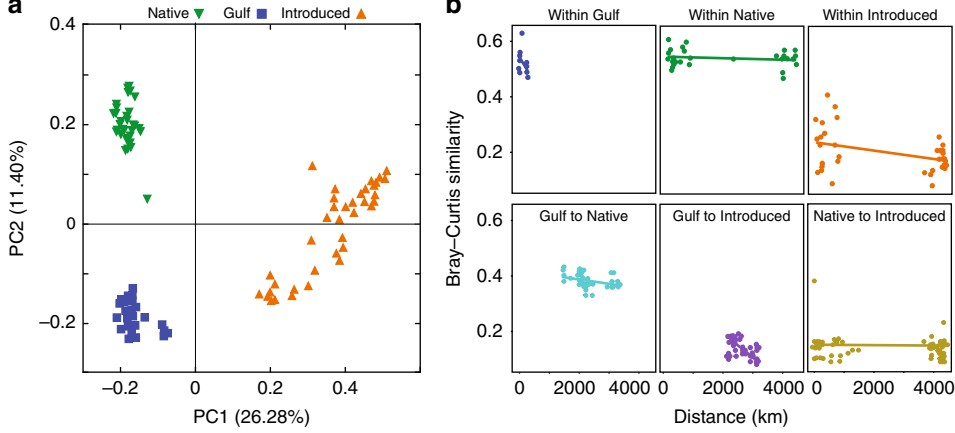

**Fig. 1** Differences in microbial community structure among lineages and as a function of distance between sampling locations. **a** Principal coordinates analysis of Bray–Curtis dissimilarity for the microbial communities associated with Native (*green*), Gulf (*blue*) and Introduced (*orange*) lineages of *Phragmites australis*, collected from across the United States (Supplementary Fig. 1). **b** Bray–Curtis dissimilarity plotted as a function of the distance (km) between sampling locations. Regression statistics are found in Supplementary Table 2. $n = 5$ replicates per *P. australis* population (eight Introduced populations, five Gulf populations, eight Native populations)

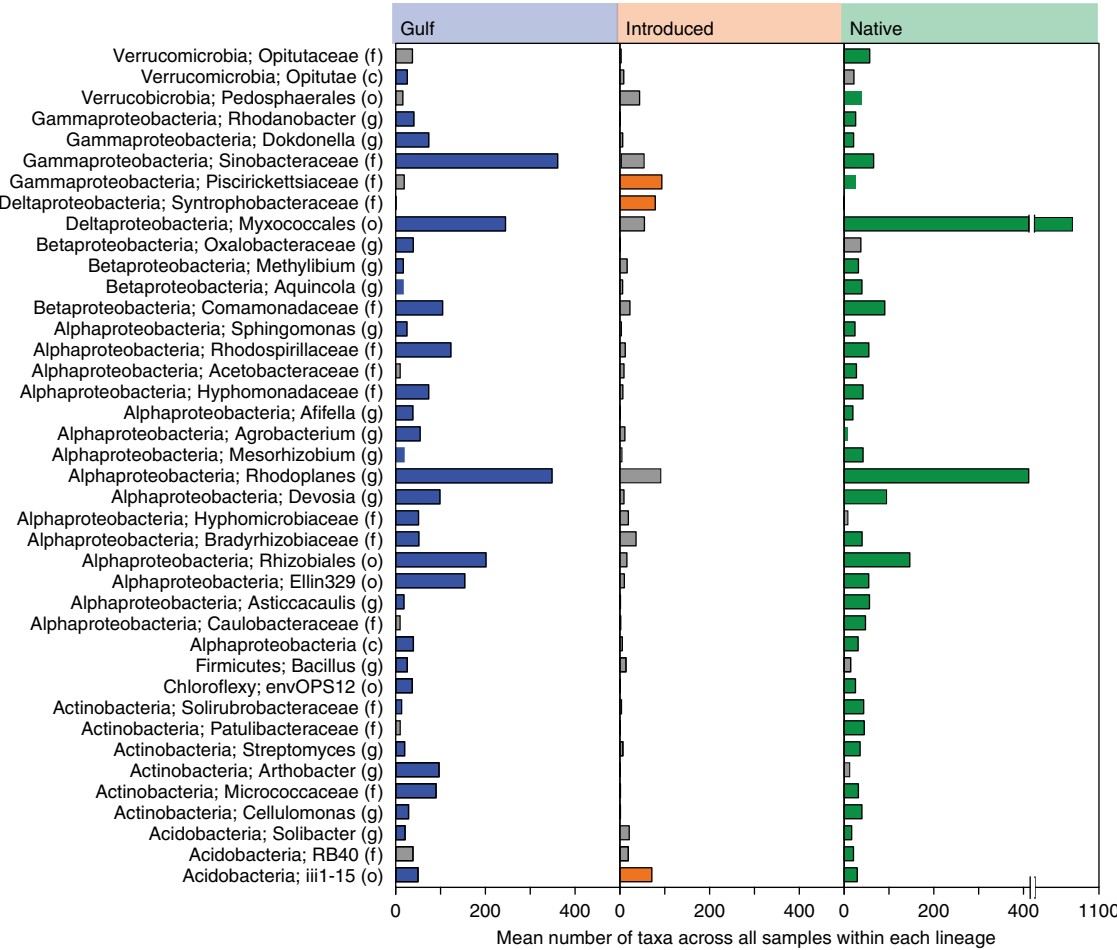

**Fig. 2** Core microbiome analysis for each lineage. Abundance of taxa belonging to the core microbiome of different *Phragmites australis* lineages. *Colored bars* indicate the taxon was present in 100% of the samples for that lineage, whereas *gray bars* indicate that it was only present in some samples

experiment, indicate that the rhizosphere bacterial community is structured by plant lineage, with the Introduced lineage demonstrating evidence of decreased enemy attack compared to the Gulf and Native lineages.

## Results

**Overview of experiments**. We performed a field survey in which we collected rhizosphere soils from multiple *P. australis* populations across three lineages at 21 locations spanning the continental United States (Supplementary Table 1, Supplementary Fig. 1). At three locations (Block Island, RI; Great Bay, NH; and Falmouth, MA), we collected rhizosphere soil samples from co-occurring Native and Introduced *P. australis* populations, allowing us to more closely examine the relative strength of both lineage and geographic proximity in structuring rhizosphere bacterial communities. It is possible, however, that homogeneity in the bacterial community within lineages could also arise from different microhabitat preferences maintained by each lineage, so we confirmed the role of lineage as a driver of bacterial community structure in a common garden experiment. We grew multiple populations from each lineage individually from surface sterilized rhizomes in homogenized (though not sterilized) potting soil under identical greenhouse conditions. For both the field and common garden experiment we used high throughput sequencing of the 16S rRNA gene to examine the entire bacterial community and its gene product, 16S rRNA, to assess which bacteria were actively synthesizing proteins and were therefore more likely to be influencing plant fitness (see Methods).

**Field survey**. Even at continental scales, our results reveal a remarkable influence of *P. australis* lineage on the structure of the bacterial community (Fig. 1a). A principal coordinates analysis of Bray–Curtis similarity indicates that the rhizosphere bacterial communities of the Native, Gulf, and Introduced lineages were distinct from one another, regardless of where in North America they were collected (permutational multivariate analysis of variance (PERMANOVA); $F_{2,107} = 22.38$, $p < 0.001$). We also found similar results when we examined the potentially active community (Supplementary Fig. 2; PERMANOVA, $F_{2,75} = 8.13$, $p < 0.001$), indicating that lineage may be a sufficiently strong determinant of bacterial community structure that it overwhelms local environmental conditions even among the active community, which would be more likely to respond to local environmental variation.

The rhizosphere bacterial communities within the Gulf and Native lineages demonstrated stronger within-lineage similarity than the Introduced lineage (Fig. 1b, Supplementary Table 2). The lower overall similarity in community structure among Introduced populations ultimately resulted in an overall higher bacterial richness in the Introduced lineage compared to the other lineages (Supplementary Fig. 3; ANOVA, $F_{5,206} = 56.5$, $p < 0.01$). Remarkably, bacterial communities associated with sympatric Native and Introduced populations were less similar to one another than populations from within the same lineage collected >3000 km apart (Fig. 1b). The similarity among the bacterial communities within the Introduced lineage decreased with distance, a pattern not observed in the Native and Gulf lineages

and indicative of a higher rate of bacterial turnover in the Introduced lineage rhizospheres.

We examined the core microbiome of each lineage (taxa present in 100% of samples collected from across the continent). Despite striking differences in community structure among lineages (Fig. 1a), bacterial taxa within lineages displayed a core set of taxa that were consistent across large geographic regions (Fig. 2). Core microbiome members were predominantly heterotrophic taxa likely able to take advantage of the high primary productivity of *P. australis*. Surprisingly, while core Native and Gulf bacteria were sometimes present in the Introduced rhizospheres, abundances were typically low (Fig. 2). For example, the deltaproteobacterial order Myxococcales was among the most abundant orders identified but 77% of all Myxococcales in the total community were found in Native rhizospheres and only 4% were found in Introduced rhizospheres (Supplementary Table 3).

**Common garden experiment**. To investigate experimentally how different *P. australis* lineages shape the rhizosphere community and whether this ecosystem engineering contributes to plant fitness, we performed a common garden experiment on a subset of the field populations (Supplementary Table 1). We also sought to tease apart the effects of lineage on plant performance as mediated by lineage-specific effects on the soil bacterial community. We hypothesized that the three lineages grown in homogenized Metromix potting soil would develop distinct rhizosphere bacterial communities, and that these communities would influence plant productivity. Although similar initially (Fig. 3a, *purple*), 4 months of conditioning by the three different lineages caused soil bacterial communities to diverge dramatically (Fig. 3, Supplementary Tables 3 and 4). The total bacterial community (Fig. 3a, *open symbols*) formed three clusters; one from the Gulf, one exclusively from Introduced, and one that was a mix of Introduced and Native. Consistent with our field results, the active bacterial community (Fig. 3b) was strongly influenced by both lineage and source population. Active bacterial communities associated with the three Native populations (Fig. 3b, *blue*) were highly similar to each other, but divergent from Introduced and Gulf communities. In contrast, each individual population of active bacteria from both the Gulf (Fig. 3b, *green*) and Introduced (Fig. 3b, *orange*) lineages formed distinct non-overlapping clusters of points that were not only different from Native populations, but also different from each other.

To assess whether the lineage-specific differences in bacterial community structure could potentially play a role in plant fitness or invasion success, we used an ancestral state reconstruction model to infer the function of the bacterial community[32]. We assessed three different KEGG pathways relevant to plant–microbe interactions, including Toxin and Metal Detoxification, Antimicrobial Biosynthesis, and Antimicrobial Degradation. All three pathways were significantly less abundant in the rhizospheres of the Introduced lineage after 4 months growing in our common garden experiment (Fig. 4).

**Structural equation modeling**. Whether changes in the bacterial community affect invasion success ultimately depends on how these changes influence plant fitness related traits, measured here as biomass. However, plant fitness might also be driven by other modified soil properties or solely by plant lineage. We teased apart these direct and indirect effects of *P. australis* lineage using Structural Equation Modeling[33–35]. Common garden Introduced populations had lower bacterial activity and metabolism than Native and Gulf *P. australis* populations. Surprisingly, this lineage effect on bacterial community structure did not translate into

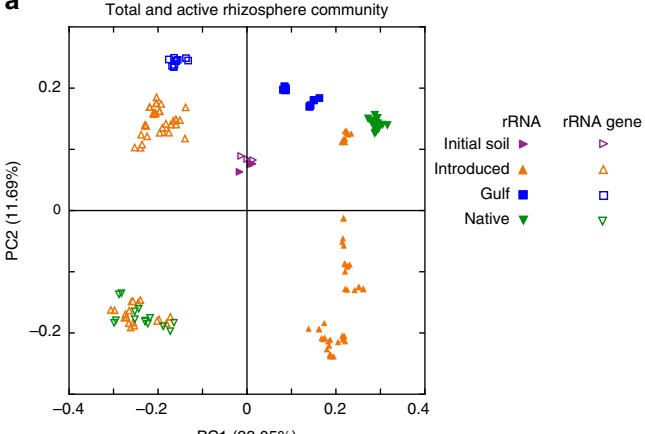

**a**

Total and active rhizosphere community

rRNA    rRNA gene
Initial soil
Introduced
Gulf
Native

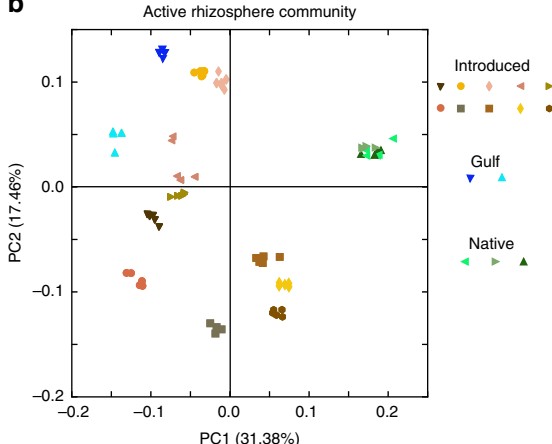

**b**

Active rhizosphere community

Introduced

Gulf

Native

**Fig. 3** Microbial community structure of different lineages of *Phragmites australis* grown in a common garden experiment. **a** Principal coordinates analysis of Bray–Curtis dissimilarity for the entire (*open symbols*) and active (*closed symbols*) microbial community, along with the pre-incubation community (*purple*). **b** Principal coordinates analysis of the Bray–Curtis dissimilarity values for just the active microbial community, indicating the importance of population (PERMANOVA, $F_{59,14} = 20.56$, $p < 0.001$), and lineage ($F_{71,2} = 19.79$, $p < 0.001$)

altered biomass of *P. australis* plants (Fig. 5, Supplementary Table 5 and 6). Rather, while *P. australis* lineage shaped the bacterial community, its activity, and belowground soil properties, the influence of lineage on aboveground and belowground biomass was solely via direct effects. Thus, *Phragmites australis* lineage appears to be the primary driver of differences in the rhizosphere ecosystem and in plant biomass, at least for the 4-month-old plants in this common garden experiment. It is worth noting, however, that the observed differences in in the rhizosphere bacterial community could have indirect effects on co-occurring plants through spillover of generalist pathogens or mutualists, or by altering competitive interactions[36], effects we did not evaluate because each *P. australis* plant was grown individually.

## Discussion

Our results expand on findings from lab-based studies of model organisms and agricultural cultivars[7–12] to document the important role of plant lineage in structuring the bacterial community of a wild plant species whose lineages diverged thousands of years ago[37, 38]. We demonstrate that different lineages of *P. australis* had unique bacterial signatures when samples were

collected from across the continental United States. Numerous studies have documented the importance of local environmental features in structuring soil microbial communities[7, 13, 28–31], thus the divergence in bacterial communities between sympatric lineages growing in the same system suggests that lineage overwhelms local environmental conditions in determining rhizosphere bacterial community structure—results we were able to confirm with a common garden experiment.

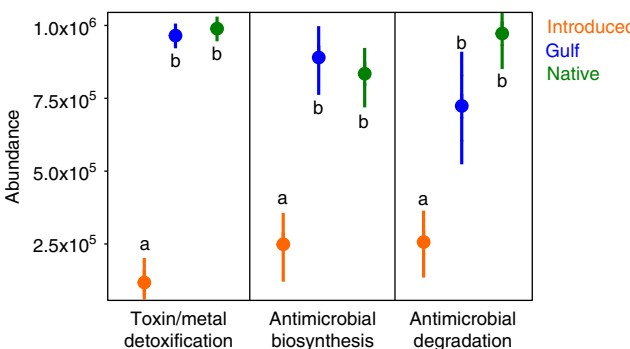

**Fig. 4** Abundance of KEGG pathways potentially associated with plant–microbe interactions in the rhizosphere soils of Introduced, Gulf, and Native lineages. In each of the three pathways, data are the mean ± standard error of the mean and different letters denote significantly different means within each pathway. The abundance of each pathway was significantly lower in the Introduced lineage rhizosphere microbial communites than in those communities associated with the Gulf and Native lineages. Toxin/metal detoxification: ANOVA, $F_{2,75} = 47.56$, $p < 0.001$; antimicrobial biosynthesis: ANOVA, $F_{2,75} = 35.23$, $p < 0.001$; antimicrobial degradation: ANOVA, $F_{2,75} = 29.33$, $p < 0.001$)

Differences between Native and Introduced *P. australis* have been observed in associated oomycete[39] and archaeal communities[40]. These consistent differences between Native and Introduced lineages across multiple microbial domains support the view that lineage exerts a dominant control on soil rhizosphere communities, one that may promote invasion success through plant–soil feedbacks, spillover, and soil legacies[36]. In our common garden experiment, we found no evidence that lineage-driven bacterial communities directly affected plant fitness in the absence of competitors. Therefore, any effects of lineage-specific bacterial communities on invasion success likely comes from effects on competitors via plant–soil feedbacks. An extensive body of literature documents the important role of plant–soil feedbacks, particularly the escape from pathogens in the native range, in promoting invasion success[15, 41, 42]. These conclusions, however, are typically inferred from measuring plant growth responses when grown in soils that have been preconditioned with either native or invasive plants[43], or from soils collected from native and invasive ranges[44]. Thus, in most cases the actual microbial mechanism for observed negative soil feedbacks is implied rather than documented. Although we have yet to test how the bacterial community changes after cross transplanting native and invasive plants into preconditioned soils, our work nonetheless sheds light on possible bacterial underpinnings of plant–soil feedbacks.

We hypothesized that if native lineages were under constant enemy attack they would have evolved specific associations with specialist rhizosphere bacteria that could aid in their defense from pathogens[15, 45, 46]. As a result, the bacterial communities associated with the native lineage should be similar regardless of geographical location. The much greater similarity observed within the Native and Gulf lineages, compared to the Introduced lineage, both in our field and our common garden experiment,

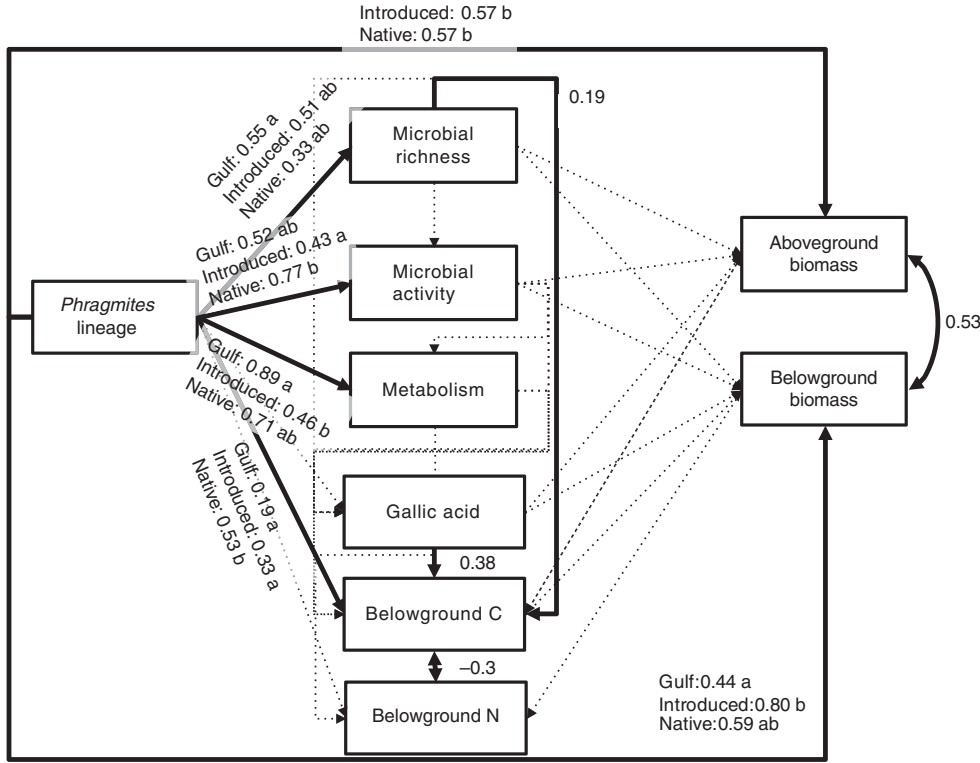

**Fig. 5** Structural Equation Model fitted with range-standardized coefficients. *Solid lines* indicate that the driver influences the likelihood of the model via a $\chi^2$ Likelihood ratio test. *Dashed paths* indicate no detectable influence of the driver ($p > 0.05$). Standardized coefficients are presented for each path. Letters *a* and *b* denote groupings via post-hoc tests. $\chi^2$ test statistics can be found in Supplementary Tables 5 and 6

supports this conclusion. The different abundances of important protein pathways that we observe (Fig. 4) add additional support for the enemy release hypothesis. Introduced *P. australis* rhizosphere microbial communities had significantly lower abundances of pathways involved in antimicrobial biosynthesis and degradation, suggesting a lower exposure to enemy attack. Introduced *P. australis*, when freed from pathogens that were present in its native range, cease to dedicate resources to cultivation of a pathogen defense system. Thus, the bacterial community in the rhizosphere soils should show a greater contribution of local soil generalists, rather than specific taxa cultivated for pathogen defense. The greater change in bacterial turnover with distance in the Introduced lineage compared to the other lineages suggests that Introduced populations have a greater reliance on novel associations in soils from the introduced range that are derived from the pool of local microflora, a pool that would be expected to decrease in similarity with distance from the source population[47].

Our results support that the *P. australis* lineage played a critical role in structuring bacterial communities across large spatial scales. The importance of lineage overwhelmed the importance of local environmental factors in structuring bacterial communities both in the field and in a common garden greenhouse experiment. Globally, the spread of invasive species has reduced biodiversity[48] and altered ecosystem function[49], and this work highlights another potential impact of species invasions—lineage-specific reshaping of endemic microbial communities. The higher belowground biomass of the Introduced *P. australis* lineage, compared to the Native lineage and to other native wetland plants it displaces, suggests that invasion-induced changes in microbial community could be widespread, though further work is needed to determine how far into the bulk soil these changes propagate. Microbes play key roles in ecosystem function and this bottom-up engineering of the rhizosphere microbial community assembly by plant lineage could have repercussions for important ecosystem services provided by microbes, such as nutrient cycling and carbon sequestration. It is therefore essential that we take the next step to determine how these lineage-specific changes in the rhizosphere bacterial community translate to changes in *P. australis* associated ecosystem function.

## Methods

**Field sampling**. During the summer and fall of 2015, we collected soil samples to assess the bacterial communities associated with Native, Introduced, and Gulf *P. australis* rhizosphere (Supplementary Table 1, Supplementary Fig. 1). We ran 20 m transects through the middle of previously identified stands of *P. australis* and excavated belowground material every 5 m, starting with an initial sampling at 0 m and collecting additional samples at 5, 10, 15 and 20 m distances (n = 5 samples per transect, which we have previously shown to be robust to capture the dynamics within and between different plant rhizosphere bacterial communities[50]). At each sampling location we collected the soil directly associated with the roots. Soil samples were preserved in RNAlater and stored at −20 °C. At the same time, aboveground plant tissue was collected to verify lineage and for measurement of carbon, nitrogen, and gallic acid concentration. Haplotypes of *P. australis* lineages were confirmed by cpDNA sequencing[23].

**Common garden experimental design**. We maintain multiple populations of *P. australis*, whose lineages have been confirmed by RFLP analysis according to the methods of Saltonstall[23], in a laboratory at the University of Rhode Island. For this experiment, we selected 15 different source populations that had been growing in the Rhode Island common garden for at least 3 years. Three source populations were from the Native lineage, two were from the Gulf lineage, and ten were from the Introduced lineage. The original plants were collected from across Eastern North America, spanning a latitudinal gradient from Quebec to Louisiana. Multiple individuals of each plant were propagated in water in the greenhouse from same length stem cuttings so replicates of each population were genetically identical and of standard size. When the aerial shoots of the propagules were sufficiently large (>2.5 cm in length) they were separated from the larger stem on which they were growing by cutting the stem 2.5 cm on either side of the node from which the stem emerged. The surfaces of the plants and roots were dipped in a 10% bleach solution to remove any surface associated microbes and each propagule was individually

planted in a clean, bleached cone-shaped "deepot" container (2.5" × 10", D40L, Stuewe and Sons, Inc.) filled with sifted and homogenized Metromix potting soil. Individual deepots were randomly positioned in racks and racks were rotated regularly to avoid location-specific effects on plant growth. Plants were grown in the greenhouse under identical conditions of water and light for 4 months, at which point individual plants were harvested. To harvest, we gently removed the entire plant from the cone. After allowing loose soil to shake free, we then collected remaining root associated soil into individual cryovials that were immediately frozen in liquid nitrogen. Remaining roots were cleaned of any residual soil and roots, shoots, and residual soils were separately bagged and dried for later analysis.

**Soil and plant tissue analysis**. Plant shoots and roots were oven-dried at 70 °C to a constant biomass and weighed to determine the total aboveground and belowground biomass for each plant. Leaves were lyophilized and ground to a fine powder in the laboratory. Carbon and nitrogen content of both the aboveground and belowground plant tissue was determined on a Perkin Elmer 2400 Series Elemental Analyzer (Waltham, MA) at the University of Massachusetts Environmental Analytical Facility following standard operating procedures. The coefficient of variation in replicate analysis was 0.016 for %C and 0.025 for %N. Total phenolics (nM g$^{-1}$ of dried plant tissue) were estimated using a microplate modified version of the Folin–Ciocalteu method[51]. Colorimetric methods were used to determine $NO_3^-$ [52]) and $NH_4^+$ [53]) concentrations using a Bio-Tek microplate reader (Powerwave 340, Winooski, VT).

**Nucleic acid extraction, amplification, and sequencing**. Five replicates from each population were haphazardly selected for molecular analysis of the total and active bacterial communities. DNA was extracted from approximately 0.25 g of soil using the MoBio PowerSoil DNA Isolation Kit (Carlsbad, CA) following manufacturer's instructions. RNA was extracted from an additional 0.5 g of soil following a protocol originally developed by Mettle et al.[54] and modified as described by Kearns et al.[50]. Briefly, cells were lysed by vortexing with 0.17 mm glass beads in 700 μl PBL buffer (5 mM Tris-HCl (pH 5.0), 5 mM $Na_2EDTA$, 0.1% (wt/vol) sodium dodecyl sulfate, and 6% (vol/vol) water-saturated phenol) for 10 min. Following centrifugation, the supernatant was transferred to a new centrifuge tube and the remaining soil and glass beads were resuspended in 700 μl TPM buffer (50 mM Tris-HCl (pH 5.0), 1.7% (wt/vol) polyvinylpyrrolidone, and 20 mM $MgCl_2$) and vortexed at maximum speed for an additional 10 min. Combined supernatant was treated with an equal volume of phenol:choloroform:isoamyl alcohol (25:24:1 v/v/v; pH 5.5) before nucleic acids were precipitated with 100% isopropanol and 3 M sodium acetate and washed with 70% ethanol. The RNA was loaded on an Illustra Autoseq G-50 Spin Column (GE Healthcare, Pittsburgh, PA) containing 500 μl prewashed Q-Sepharose (GE Healthcare). Samples were spun at 650 × g and eluted five times with 80 μl of 1.5 M NaCl (pH 5.5). The eluate was again precipitated with 100% isopropanol and 3 M sodium acetate, and washed with 70% ethanol before being eluted into 50 μl di-ethyl pyrocarbonate treated water. Any DNA contamination in the RNA was removed with DNase I (New England Biolabs, Ipswich, MA) following manufacturer's instructions and RNA was reverse transcribed to cDNA with random hexamer primers using the Invitrogen Superscript RT III cDNA synthesis kit (Grand Island, NY). cDNA synthesis was checked by amplification with general bacterial primers and products were visualized via gel electrophoresis on an ethidium bromide stained 1.5% agarose gel.

DNA and cDNA from each sample was quantified via Qubit 2.0 (ThermoFisher, Waltham, MA, USA) and Picogreen (Invitrogen, Grand Island, NY) and all samples were normalized to a nucleic acid concentration of 3 ng μl$^{-1}$ for subsequent PCR. We amplified a 291 base pair fragment of the V4 region of the 16S rRNA and 16S rRNA gene using primers 515 F and 806R[55] that were adapted for sequencing[56] on an Illumina MiSeq (San Diego, CA). Each sample was amplified in triplicate with a unique 12 base barcode to enable pooling of multiple samples per sequencing run. PCR products were cleaned via gel purification using the Qiagen QIAquick gel extraction kit (Valencia, CA) and were fluorometrically quantified using a ThermoFisher Qubit (Waltham, MA). Final PCR products were pooled in equimolar ratios and sequenced using the Illumina MiSeq platform with a 300 cycle V2 chemistry sequencing kit.

**Data analysis**. After sequencing, fastq-join[57] was used to match paired end reads using default parameters and the resulting data were demultiplexed and quality filtered[58] in QIIME[59] (version 1.9.1). We used closed-reference picking with uclust[60] to determine operational taxonomic units (OTUs) at a 97% sequence identity using the Greengenes 13.8 Core Reference Alignment[61]. Chimeric sequences were identified and removed using Chimera Slayer[62] and singletons were also removed prior to downstream analysis. We compared the bacterial community composition of each sample by calculating the Bray–Curtis dissimilarity metric and we visualized the resulting distance metric using a principal coordinates analysis in QIIME. Alpha diversity was calculated based on data rarified to the lowest sequencing depth, 5394 sequences per sample. We calculated a core microbiome, defined as taxa that were present in 100% of the samples within a given lineage using QIIME, and we assessed significant differences in taxonomic abundance among the different lineages using a Kruskal–Wallis test followed by a Benjamini–Hochberg correction to correct for false discovery.

We assessed differences in bacterial community structure among Native, Gulf, and Introduced *P. australis* lineages using Adonis, which is a PERMANOVA[63] implemented in QIIME, with significance assessed at an alpha of 0.01. We assessed changes in community structure with distance by plotting the pairwise Bray–Curtis dissimilarity coefficient against pairwise distance between locations and calculated the linear regression between the two in R. We used PiCrust[64], which assigns function based on 16S rRNA marker sequences using an ancestral state reconstruction approach, to determine bacterial metabolisms. To assess differences in potential bacterial metabolisms among lineages, we implemented PiCrust on Galaxy[64–66], using closed-reference OTU picking in QIIME. The resulting biom file was uploaded to Galaxy (http://huttenhower.sph.harvard.edu/galaxy/) and data were normalized to account for multiple copies of 16S rRNA in some bacteria. After normalization PiCrust predicted the metagenomic content of each sample. We then used HUMAnN[67] to generate pathway summaries and separated out those pathways associated with bacterial metabolisms for use in the structural equation model. We tested for significant differences in three specific pathway associated with plant–microbe interactions, toxin and metal detoxification, antimicrobial biosynthesis, and antimicrobial degradation using analysis of variance, followed by Tukey's post-hoc test in R.

**Structural equation modeling**. We used mixed model Structural Equation Modeling (SEM)[45–47] to assess the interactions among *P. australis* lineage, bacterial processes, and plant and soil properties. All data and code for the SEM is available at: at https://github.com/jebyrnes/phrag_common_garden_sem. Parameters in the model included *P. australis* lineage, above and belowground plant biomass, belowground gallic acid concentration (phenolics), soil percents and nitrogen, and three bacterial parameters: number of OTUs, activity (per taxon ratio of 16S rRNA to the 16S rRNA gene), and metabolism (calculated from KEGG pathways indicated by PiCrist). We allowed *P. australis* lineage to affect bacterial richness (number of taxa), activity (16S rRNA:16S rRNA gene ratio), and metabolism (based on ancestral state reconstruction of metabolic pathways). Lineage and all ecosystem variables could affect aboveground and belowground biomass, and all variables influenced soil carbon and nitrogen pools, which could also covary.

We fit and evaluated the model using restricted maximum likelihood with the *nlme* and *piecewiseSEM* package[47, 68] in R (version 3.31). Given that lineage was categorical, we evaluated whether different variables influenced one another using $\chi^2$ Likelihood ratio tests for individual model pieces. We then report the estimated path coefficient. We compared individual lineage effects using post-hoc means adjusted Tukey tests[69]. Exploration of non-additive models with *P. australis* lineages interacting with different endogenous variables showed no nonlinearities.

**Data availability**. All sequences associated with this study are available from the Sequence Read Archive under accession numbers SRR4419841 (greenhouse associated sequences) and SRR4420130 (field collected sequences). All data and code associated with the structural equation modeling are available at https://github.com/jebyrnes/phrag_common_garden_sem.

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

## Acknowledgements

We thank Petr Pyšek, A. Randall Hughes and José Amador for comments on earlier versions of this manuscript. Funding was provided by NSF grants to J.L.B. (DEB1350491), L.A.M. (DEB1049914), J.T.C. (DEB1050084), and W.J.A. (DEB1501775). Additional support was provided by the University of Rhode Island to L.A.M. (RI00H-332, 311000-6044), the Louisiana Environmental Education Commission to W.J.A. and MIT Sea Grant to J.E.K.B. (2014-R/RCM-36). We thank Rose Martin and Serena Moseman-Valtierra for early discussions on this project and Jeff Dusenberry and the Research Computing Center at University of Massachusetts Boston.

## Author contributions

J.L.B., L.A.M., W.J.A. and J.T.C. designed the experiments, J.E.K.B. performed structural equation modeling, S.W. and W.J.A. collected field samples, L.A.M. and S.W. maintained the greenhouse experiment, P.J.K., M.G., K.T., S.W. and J.Y. performed all sequence processing and analysis under the guidance of J.L.B., J.T.C. provided logistical support for field operations, J.L.B., L.A.M., W.J.A., J.T.C., J.E.K.B. and P.J.K. wrote the manuscript.

## Additional information

**Competing interests:** The authors declare no competing financial interests.

