## [Peer Review File · Nature Communications]

Reviewers' comments:

Reviewer #1 (Remarks to the Author):

The manuscript by Bowen et al describes an interesting study on the lineage specific impact of *Phragmites australis* lineages on the structure of bacterial communities in the rhizosphere.

Although the title suggests that the total rhizomicrobiome is considered the study is limited to bacterial communities only; for instance fungi are not included and I do not know to what extent the archaeal community is covered. This is repeated throughout the entire manuscript and I suggest adjusting this in a new version.

The data clearly indicate that lineage is a strong determinant of bacterial communities in the rhizosphere. Although interesting, it is not a novel observation. Already from the early days of modern rhizosphere research, numerous studies have been directed towards the relative importance of soil/edaphic factors versus plant species/cultivar in shaping the microbial community structure. In most cases soil/edaphic factors were found to influence microbial communities in the rhizosphere more than do plant related factors, but the opposite was found as well. Some of these studies also observed significant cultivar effects. Although most of these studies were performed using crops and crop cultivars, in this case of rhizosphere effects, I do not consider this to be essentially different from studies on wild plant lineages. Also the distance related diversity patterns have been observed previously although the observation of differences in distance related diversity decay between the introduced and native/gulf lineages is, as far as I know, novel.

Samples were taken from a wide range of locations throughout the USA. The paper suggests that this sampling strategy implies that the variation of edaphic and environmental conditions under which the plants were growing and the samples were taken was spread evenly over the lineages. However, the information provided does not allow for such conclusion. As the authors state correctly plant-soil/rhizosphere effects are strongly localized. So, it could well be that lineages were growing at (and so samples were taken from) specific local environmental /edaphic conditions, which would mean that the lineage effects considered here are in fact environmental/edaphic effects. The data on the sampling sites do not provide information on key factors such as pH, water conditions, temperature, etc. Factors such as pH and nutrient content can vary over very short distances, which may therefore also explain the observations on the different microbial communities in adjacent stands. So, without this information on the key environmental factors of the sample sites it is not possible to conclude "that lineage is a sufficiently strong determinant of microbial community structure that it overwhelms local environmental conditions"(line 81-83).

The field sampling design is not clear to me. Line 166-170 states that over a transect of 20 meters each 5 meters a sample was taken, resulting in 4 samples per site, isn't? However line 206-207 states that "Five replicates from each population were haphazardly selected for molecular analysis of the total and active microbial communities ", which is not equal to the former statement . It may be that this refers to the common garden experiment only, but then more information on the field sampling is needed; if it does not refer to the common garden experiment only, but also refers to the field sampling, how is this then related to the former statement?

Given the observations of the common garden experiment, the conclusions provided in lines 156-164 on the role of the microbial community in the ecological performance of invasive species are rather overstated and should be reconsidered. Also the impact of invasive species on the endemic microbial communities is somewhat overdone. If a plant species attracts specific groups of microbes in its rhizosphere it does not mean that the total soil community is equally affected. So, more realistic concluding statements should be formulated in line with the observations described.

Also the statements in lines 116-124 on the potential role of Myxococcales in native and

introduced populations of *P.austrarlis* are rather speculative. The list of organisms found here contains many antibacterial compound producing species and some had similar abundances. So either all most abundance species are discussed in more detail or this section should be reconsidered and reformulated.

I liked the use of the Structural Equation Modeling approach. It nicely shows the relative importance of the different factors involved. I wondered what the authors tested in case of soil C and N. Figure 4 suggests that belowground C and belowground N were considered separately, but the text, line 270, indicates that the soil C:N ratio was included. Although related, C:N ratio is mathematically different from soil C and soil N. So, the question rises what was tested.

Overall, my conclusion is that this is an interesting paper but it needs further reconsideration and additional information and also due to its lack of novelty, I do not consider it acceptable for Nature Communication

Reviewer #2 (Remarks to the Author):

This interesting and well-written paper clearly demonstrates that different lineages of *Phragmites australis* cultivate different microbial symbionts, even from very different soil sources. To my knowledge, this level of evidence for genomic control of microbial symbionts in plants fills an important knowledge gap. However, the gap might be a little smaller than presented in the paper. Maron et al. (2015) reported evidence for large differences among populations (from both the native and non-native range) of *Solidago gigantea* in the strengths of the plant-soil feedbacks. This study found that both intraspecific variation and geographical soil sources contributed to variation in feedback intensity. This species is native to North America and invasive in Europe. To be clear, your genomic evidence takes your study to a different level, and the results are very interesting, but the Maron paper (and citations within) are quite relevant because of the population-level variation in the strengths of feedbacks.

In your results, I can ascertain clearly the effects of lineage on the composition of the microbial symbiont community. However, after multiple readings I am not as clear on the "plant-soil feedback" aspect of your study. To my knowledge, studies of such feedbacks grow plants in soil for a time to "train" or "condition" that soil, and then use various approaches (filters, sterile and non-sterile aliquots, etc.) to examine whether the trained soil affects the growth of a conspecific or another species. Or at least there is some general version of this approach. I could not tell clearly, what you did to actually generate and measure feedbacks, or even if you actually measured feedbacks in some sort of typical or classic way. This might simply require a more careful and detailed presentation in the methods of how you got the actual feedbacks, or more importantly, perhaps you will have to reconsider the strong focus on "feedbacks" (if you haven't really measured them) and instead put your work more strongly in the context of lineage effects on the microbiome. This is important to clarify and figure out because you have very strongly developed the paper in the context of plant-soil feedbacks, and report results in the context of feedbacks.

Maron, J.L., et al. *Oecologia* (2015) 179: 447. doi:10.1007/s00442-015-3350-2

Reviewer #3 (Remarks to the Author):

The manuscript of Bowen et al shows that plant lineage is a stronger selector of rhizobacteria than

plant biogeography.

The question is important by an ecological point of view. The research is mostly done under a molecular microbiology point of view, that maybe is a bit outside the interdisciplinarity requested for these kind of researches.

Although the experimental design could be read as generally adequate, to my point of view there are some points to be further investigated before the acceptance.

1- The importance of the effects of lineage or cultivars on the rhizobacteria, stronger than other parameters, is already assessed in different papers, mostly from the agricultural or GMO sides; For instance, Brusetti et al, *Plant Soil* 2005; Bouffaud et al *Mol Ecol* 2012; Wen et al, *Biol Fertil Soils* 2016; Aira et al, *Soil Biol Biochem* 2010.... Which are the real differences between these results and those of the other scientists?

2- The analyses lack of correlation with environmental data, such as water flux (if any), soil/sediment chemistry than maybe could explain part of the observed results.

3- Are these differences, looked even at the RNA level, a mirror of functional, physiological and biochemical changes into the local microbial communities? This is an important task to be investigated.

We would like to thank the reviewers for their thoughtful comments on our manuscript. We have addressed each of their comments in the manuscript as described below in italics.

Reviewers' comments:

Reviewer #1 (Remarks to the Author):

The manuscript by Bowen et al describes an interesting study on the lineage specific impact of *Phragmites australis* lineages on the structure of bacterial communities in the rhizosphere. Although the title suggests that the total rhizomicrobiome is considered the study is limited to bacterial communities only; for instance, fungi are not included and I do not know to what extent the archaeal community is covered. This is repeated throughout the entire manuscript and I suggest adjusting this in a new version.

We changed the word 'microbial' to 'bacterial' throughout the manuscript, including the title, except where we were speaking about generalizable patterns, rather than the specifics of this study.

The data clearly indicate that lineage is a strong determinant of bacterial communities in the rhizosphere. Although interesting, it is not a novel observation. Already from the early days of modern rhizosphere research, numerous studies have been directed towards the relative importance of soil/edaphic factors versus plant species/cultivar in shaping the microbial community structure. In most cases soil/edaphic factors were found to influence microbial communities in the rhizosphere more than do plant related factors, but the opposite was found as well. Some of these studies also observed significant cultivar effects. Although most of these studies were performed using crops and crop cultivars, in this case of rhizosphere effects, I do not consider this to be essentially different from studies on wild plant lineages. Also the distance related diversity patterns have been observed previously although the observation of differences in distance related diversity decay between the introduced and native/gulf lineages is, as far as I know, novel.

We have added a paragraph that summarizes more of the cultivar literature detailing plant soil feedbacks on lines 59-69. We also modified our title, and text (lines 53-55, 67-69, and 101-133) to emphasize the novel aspects of this contribution, including that we examine both genotype and geographical location, that we are focused on natural instead of agricultural systems, and that we examined both the total and active community of bacteria.

Samples were taken from a wide range of locations throughout the USA. The paper suggests that this sampling strategy implies that the variation of edaphic and environmental conditions under which the plants were growing and the samples were taken was spread evenly over the lineages. However, the information provided does not allow for such conclusion. As the authors state correctly plant-soil/rhizosphere effects are strongly localized. So, it could well be that lineages were growing at (and so samples were taken from) specific local environmental /edaphic conditions, which would mean that the lineage effects considered here are in fact environmental/edaphic effects. The data on the sampling sites do not provide information on key factors such as pH, water conditions, temperature, etc. Factors such as pH and nutrient content can vary over very short distances, which may therefore also explain the observations on the different microbial communities in adjacent stands. So, without this information on the key environmental factors of the sample sites it is not possible to conclude "that lineage is a sufficiently strong determinant of microbial community structure that it overwhelms local

environmental conditions” (line 81-83).

We agree with the reviewer that it is possible that the optimal habitats for each lineage could be slightly different, thus confounding the lineage by environment pattern we observe. This is why we also performed the common garden experiment, growing all lineages in a common soil, to confirm that the effect that we observed was due specifically to lineage effects and not to the local soil environment. We have clarified our reasoning in this regard on lines 148-153, and softened the language to say that lineage ‘may be’ a sufficiently strong determinant...”on lines 186-187.

The field sampling design is not clear to me. Line 166-170 states that over a transect of 20 meters each 5 meters a sample was taken, resulting in 4 samples per site, isn't? However line 206-207 states that “Five replicates from each population were haphazardly selected for molecular analysis of the total and active microbial communities”, which is not equal to the former statement . It may be that this refers to the common garden experiment only, but then more information on the field sampling is needed; if it does not refer to the common garden experiment only, but also refers to the field sampling, how is this then related to the former statement?

We began each transect and first sampling location at 0 meters, and collected at five meter increments for 20 meters (0, 5, 10, 15, 20 = 5 samples). We clarified this in the text on lines 473-476.

Given the observations of the common garden experiment, the conclusions provided in lines 156-164 on the role of the microbial community in the ecological performance of invasive species are rather overstated and should be reconsidered. Also the impact of invasive species on the endemic microbial communities is somewhat overdone. If a plant species attracts specific groups of microbes in its rhizosphere it does not mean that the total soil community is equally affected. So, more realistic concluding statements should be formulated in line with the observations described.

We have modified the text to remove reference to the effects of microbes on the plant communities. The reviewer is correct that we have not demonstrated that effect exists. Instead we have focused on the ways that these changes in microbial communities could affect ecosystem services provided by microbes, including nutrient cycling and carbon sequestration. These changes can be found on lines 425-428. We also amended the text to describe the areal extent of the rhizosphere to provide a context for how large an area would be affected by shifts in the bacterial community. This change can be found on line 430-423.

Also the statements in lines 116-124 on the potential role of Myxococcales in native and introduced populations of *P.australis* are rather speculative. The list of organisms found here contains many antibacterial compound producing species and some had similar abundances. So either all most abundance species are discussed in more detail or this section should be reconsidered and reformulated.

We minimized the discussion of the Myxococcales with regard to its potential role in antibiotic resistance but we do still feel it is worth calling out specifically as it is the most abundant taxon overall that we found and was dramatically more abundant in the Native lineage. However, we supplemented this analysis by examining, using ancestral state reconstruction, the likely protein pathways of the bacterial communities associated with each lineage and show that the

Introduced lineage has a significantly lower abundance of the pathways responsible for antimicrobial biosynthesis and degradation as well as toxin and metal detoxification pathways when compared to the Native and Gulf lineages. We have added the results of this analysis as Fig. 4, described those results on lines 318-324, and discussed the relevance of this to our understanding on lines 402-408.

I liked the use of the Structural Equation Modeling approach. It nicely shows the relative importance of the different factors involved. I wondered what the authors tested in case of soil C and N. Figure 4 suggests that belowground C and belowground N were considered separately, but the text, line 270, indicates that the soil C:N ratio was included. Although related, C:N ratio is mathematically different from soil C and soil N. So, the question rises what was tested.

Thank you for catching this – it should have indicated that % carbon and % nitrogen were including in the model, not the ratio. We have corrected this on line 624.

Overall, my conclusion is that this is an interesting paper but it needs further reconsideration and additional information and also due to its lack of novelty, I do not consider it acceptable for Nature Communication

We would like to thank the reviewer for constructive comments that helped improve the quality of this manuscript. We hope that by highlighting more robustly the novel aspects of this work via enhanced discussion of the lineage specific distance decay response, the different abundances of putative protein pathways, that we examine both total and active microbial communities, and that we are focused on an uncultivated invasive species, we have convinced the reviewer of the novelty of our findings.

Reviewer #2 (Remarks to the Author):

This interesting and well-written paper clearly demonstrates that different lineages of *Phragmites australis* cultivate different microbial symbionts, even from very different soil sources. To my knowledge, this level of evidence for genomic control of microbial symbionts in plants fills an important knowledge gap. However, the gap might be a little smaller than presented in the paper. Maron et al. (2015) reported evidence for large differences among populations (from both the native and non-native range) of *Solidago gigantea* in the strengths of the plant-soil feedbacks. This study found that both intraspecific variation and geographical soil sources contributed to variation in feedback intensity. This species is native to North America and invasive in Europe. To be clear, your genomic evidence takes your study to a different level, and the results are very interesting, but the Maron paper (and citations within) are quite relevant because of the population-level variation in the strengths of feedbacks.

We agree with the reviewer and have expanded our discussion to include a section on how adding the genomic data enhance our understanding of plant – microbe interactions in the context of studies that have shown changes in the microbial community structure among lineages and studies that have shown lineage specific changes in plant-soil feedbacks – including the Maron paper. This discussion can be found on lines 355-396.

In your results, I can ascertain clearly the effects of lineage on the composition of the microbial symbiont community. However, after multiple readings I am not as clear on the “plant-soil

feedback” aspect of your study. To my knowledge, studies of such feedbacks grow plants in soil for a time to “train” or “condition” that soil, and then use various approaches (filters, sterile and non-sterile aliquots, etc.) to examine whether the trained soil affects the growth of a conspecific or another species. Or at least there is some general version of this approach. I could not tell clearly, what you did to actually generate and measure feedbacks, or even if you actually measured feedbacks in some sort of typical or classic way. This might simply require a more careful and detailed presentation in the methods of how you got the actual feedbacks, or more importantly, perhaps you will have to reconsider the strong focus on “feedbacks” (if you haven’t really measured them) and instead put your work more strongly in the context of lineage effects on the microbiome. This is important to clarify and figure out because you have very strongly developed the paper in the context of plant-soil feedbacks, and report results in the context of feedbacks.

The reviewer is correct. We did not perform a formal plant-soil feedback experiment of the type described. We added paragraphs on lines 355-413 to the discussion that highlights how this study fits in to the broader plant-soil feedback literature.

Maron, J.L., et al. *Oecologia* (2015) 179: 447. doi:10.1007/s00442-015-3350-2

We thank the reviewer for pointing us in the direction of this reference and for the other constructive comments that improved the quality of our manuscript.

Reviewer #3 (Remarks to the Author):

The manuscript of Bowen et al shows that plant lineage is a stronger selector of rhizobacteria than plant biogeography. The question is important by an ecological point of view. The research is mostly done under a molecular microbiology point of view, that maybe is a bit outside the interdisciplinarity requested for these kind of researches. Although the experimental design could be read as generally adequate, to my point of view there are some points to be further investigated before the acceptance.

1- The importance of the effects of lineage or cultivars on the rhizobacteria, stronger than other parameters, is already assessed in different papers, mostly from the agricultural or GMO sides; For instance, Brusetti et al, *Plant Soil* 2005; Bouffaud et al *Mol Ecol* 2012; Wen et al, *Biol Fertil Soils* 2016; Aira et al, *Soil Biol Biochem* 2010.... Which are the real differences between these results and those of the other scientists?

We thank the reviewer for indicating these references and have added a section to the introduction in which we describe their findings and also indicate the novel contribution of our work. This can be found on lines 59-91.

2- The analyses lack of correlation with environmental data, such as water flux (if any), soil/sediment chemistry than maybe could explain part of the observed results.

We agree, and as pointed out by reviewer one as well, the environmental conditions that are specific to each lineage could confound our field results, which is why we also performed the

common garden experiment. We added a sentence to clarify this point on lines 148-153.

3- Are these differences, looked even at the RNA level, a mirror of functional, physiological and biochemical changes into the local microbial communities? This is an important task to be investigated.

It is a function of the microbes that have enough resources (carbon supply, oxidant, etc) to actively grow, compared to other bacteria that may be dormant or recently dead.

REVIEWERS' COMMENTS:

Reviewer #1 (Remarks to the Author):

The response of the authors to my comments is adequate. They have considered the comments seriously and the changes of the text according to the commentary are sound and well thought out. In this form I consider the manuscript to be acceptable for publication

Reviewer #2 (Remarks to the Author):

[No further comments for author.]